# The Physical and Mental Well-Being of Medical Doctors in the Silesian Voivodeship

**DOI:** 10.3390/ijerph192013410

**Published:** 2022-10-17

**Authors:** Ewa Niewiadomska, Beata Łabuz-Roszak, Piotr Pawłowski, Agata Wypych-Ślusarska

**Affiliations:** 1Department of Biostatistics, Faculty of Health Sciences in Bytom, Medical University of Silesia, 40-055 Katowice, Poland; 2Department of Neurology, Institute of Medical Sciences, University of Opole, 45-040 Opole, Poland; 3Department of Neurological Rehabilitation, Upper Silesian Rehabilitation Center “Repty”, 57-126 Tarnowskie Góry, Poland; 4Department of Epidemiology, Faculty of Health Sciences in Bytom, Medical University of Silesia, 40-055 Katowice, Poland

**Keywords:** health condition, anxiety disorders, depressive disorders, medical doctors, 2018

## Abstract

Physical and mental well-being are important determinants of medical doctors’ lives, including their professional activity, which has a direct impact on the health and lives of patients. The aim of the study was to assess the health condition of medical doctors from the Silesian Voivodeship. The physical health condition, including the prevalence of chronic diseases, drug therapy, and pro-health activities, was assessed. Data on mental health according to the HADS scale (hospital anxiety and depression scale) referred to the selected socio-economic and professional aspects of life, as well as life and job satisfaction. The doctors most often reported diseases of the circulatory system and diseases of endocrine system, nutritional status and metabolic changes, allergies, and degenerative changes in the musculoskeletal system and connective tissue. Chronic diseases and anxiety or depressive disorders affected doctors aged 50–80 years more frequently, and those doctors undertook preventive pro-health activities more often. The higher risks of anxiety and depression were related to their social and professional roles, lower economic status, overweight or obesity, chronic diseases, pharmacotherapy, diets, and chronic fatigue. On the other hand, frequent physical activity, a longer sleep duration, smaller weekly workload, type of rest, a higher sense of job satisfaction, and a higher level of life satisfaction reduced the risk of disorders. The health condition of medical doctors in the Silesian Voivodeship requires decisive remedial actions in the professional and non-professional spheres.

## 1. Introduction

Physical well-being and mental well-being, being closely related, are important determinants of medical doctors’ lives, including their professional activity, which, in the medical profession, has a direct impact on the health and lives of patients. Data on the health status of health care and social assistance professionals indicated that these professional groups were highly exposed to the occurrence of undesirable health consequences, including the occupational diseases. On the basis of the official register of 2016, the occupational disease incidence rate in Poland was at the level of 17.7/100,000 employees, and it was 24% higher than the national one [1]. According to the data of the Central Register of Occupational Diseases in 2009–2016, in Poland, 13.8% of patients (due to their profession) were dentists, while every tenth patient was a doctor (10.7%) [1]. Among doctors, infectious or parasitic diseases were the most frequently diagnosed diseases (76.6%), while among dentists, diseases of the peripheral nervous system and the locomotor system were predominant (41%). Musculoskeletal disorders were diagnosed among dentists 47 times more frequently than among doctors and 58 times more frequently than among nurses. The age of the patients was also taken into account. For doctors, the mean age was 50.2 ± 12.1 years, while for dentists it was 58.4 ± 7.6 years [1]. According to the report of the Central Register of Doctors from December 2021, it should be noted that people of pre-retirement age (51–65 years old) comprised over 30%, and elderly people aged 65+ comprised over 20%, of all professionally active doctors and dentists in Poland [2]. These people are particularly disposed to chronic diseases. In addition, the WHO draws attention to the health effects of occupational hazards in the health sector, including exposure to infections (tuberculosis, hepatitis B and C, HIV/AIDS, and respiratory diseases), patient handing (back injury, chronic back pain), chemicals (asthma), radiation (skin and blood damage, cataracts, infertility, birth defects, and cancer), stress (burnout, chronic fatigue), and noise (annoyance, lack of sleep) [3].

For doctors, not only age but also chronic stress [4,5] may affect the development of cardiovascular diseases, obesity, and depression [6]. According to the meta-analysis results based on 31 cross-sectional studies (9447 individuals) and 23 longitudinal studies (8113 individuals), the prevalence of depression or depressive symptoms among medical doctors is at the level of 28.8% (95%CI: 25.3%–32.5%) [7]. The medical profession is associated with a high sense of responsibility for health and human life. It is accompanied by constant tension, the requirement of constant vigilance, and the pressure associated with performing tasks that have to be carried out within a certain time [8]. This profession is exposed to numerous specific factors, including strong emotions and tensions, daily contact with death, an awareness of insufficient knowledge, organization and working conditions, interpersonal conflicts, and legal responsibility [9]. In extreme situations, excessive exposure to stressful situations at work, chronic physical fatigue, and individual health may affect the occurrence of occupational burnout, i.e., the erosion of work engagement [10]. It should be noted that the burnout rate among medical doctors, according to the Medscape National Physician Burnout and Suicide Report, increased from the level of 39.8% in 2013 to 46.0% in 2015 and 43% in 2020 [11].

The present review of the literature shows that, in the analysis of the health condition of doctors, only the issues regarding occupational exposure, especially the exposure to chronic stress, are taken into account [1,4,5,6,7,8,9,10,11]. However, there are no studies on the general health profile of Polish doctors. Therefore, the aim of the study was to assess the physical and mental health of doctors in the Silesian Voivodeship and their pro-health activities in relation to selected socio-economic and professional aspects, as well as life and job satisfaction. This aim included the assessment of: 1. the frequency of chronic diseases, the use of drug therapy, and pro-health measures; 2. the occurrence of anxiety symptoms (HADS-A) and depression symptoms (HADS-D); and 3. the relationship between the occurrence of anxiety/depressive symptoms and selected socio-economic and professional aspects of life, as well as life and job satisfaction.

## 2. Materials and Methods

### 2.1. Characteristics of the Study Group

A cross-sectional study was conducted based on a group of 701 medical doctors in the period from January to December 2018. The study group included 336 (47.9%) women and 365 (52.1%) men aged 25 to 80 (mean 43.1 ± 11.8 years). Due to the age structure of the doctor population, the results were presented in two age groups: 25–50, including individuals who were 25 years old or more and less than 50 years old (N = 492; 70.2%), and 50–80, including individuals who were 50 years old or more and less than 80 years old (N = 209; 29.8%) [12]. The vast majority of the respondents were employed in a hospital (*n* = 530; 75.6%) and/or in an outpatient clinic (*n* = 544; 77.6%), and 317 people (45.2%) declared having a business as a part of a medical practice. The period of professional work of the surveyed doctors ranged from 1 to 55 years (mean 17.5 ± 11.7 years), while the number of working hours ranged from 1 to 100 h per week (mean 52.7 ± 11.3 years). The respondents (*n* = 596; 85%) represented 30 specialities in the fields of medicine and dentistry, most often internal diseases (22.8%), paediatrics (15.8%), family medicine (13.4%), general surgery (8.1%), obstetrics and gynaecology (6.4%), and laryngology (5.1%).

### 2.2. Eligibility Criteria

A total of 701 participants were enrolled from the Silesian Voivodeship in Poland. The study used a cluster selection method. The quantitative compliance of individual medical and dental specializations with the Polish population, taking into account the 2018 register of the Supreme Medical Chamber, was ensured in the study. The criteria for inclusion were: 1. the right to practice as a doctor and/or a dentist, and membership of the Silesian Medical Chambers (Czestochowska, Slaska, Bielska); and 2. consent to participate in the study, expressed by filling in a questionnaire. The minimum size of the tested sample was 378, with the confidence level of 95% and the permissible error of 5%. Assuming the uncertainty margin, 1000 questionnaires were issued, with a return rate of 70.1%.

The study was not a medical experiment, and the collected data was fully anonymous. Therefore, the consent of the Bioethics Committee was not required (protocol code: KNW/0022/KB/90/19).

### 2.3. Research Tool

The data on the health status and preventive health care measures, as well as the sociodemographic, economic, professional, and lifestyle data, were collected through a questionnaire using the PAPI (paper and pencil interview) rule. Moreover, mental well-being was assessed in the context of anxiety and/or depressive symptoms based on the HADS (hospital anxiety and depression scale). The questionnaire contained 7 statements about anxiety (HADS-A) and 7 statements about depression (HADS-D), assessed on a scale from 0 to 3 points. Based on the sum of the points, one can determine a normative and boundary status for a total number of points below 12, or the presence of anxiety/depressive symptoms for 12 or more points [13]. The results were also compared to the level of satisfaction with life according to the SWLS scale (satisfaction with life scale—5 statements with a scale from 0 points, ‘completely disagree’, to 7 points, ‘completely agree’) [14] and the level of professional satisfaction (17 statements with a scale from 0 points, ‘very dissatisfied’, to 6 points, ‘very satisfied’) [15], expressed by the total number of points obtained in a given questionnaire.

### 2.4. Statistical Analyses

Statistical calculations were performed using Excel 2019 (Microsoft) and STATISTICA v.13.3 (Stat Soft Polska). The measurable data were characterized using the mean X and standard deviation S, as well as the median M and quartile range IQR. The compliance of the distribution of the variables with the normal distribution was verified using the Shapiro–Wilk test. The significance of the mean differences was verified with Student’s *t*-test or the Mann–Whitney U test. For nominal data, the quantitative percentage notation and the X2 test or the Fisher test were used. Additionally, in order to assess the relationship between the occurrence of anxiety/depressive symptoms and the selected factors, crude odds ratios were determined with the 95% confidence interval. The level of *p* < 0.05 was taken as the criterion for statistical significance.

## 3. Results

The chronic diseases affected 480 (68.5%) of the surveyed doctors, including 284 people aged 25–50 and 196 people aged 50–80 (57.7% vs. 93.8%, *p* < 0.0001). The most frequently mentioned health problems are presented at Figure 1. Other chronic diseases (reported by less than 1% of respondents) included neoplasms and respiratory, cardiovascular, and digestive diseases, as well as anxiety and depression.

Moreover, 464 (66.2%) of the surveyed doctors declared constant use of drugs, including antiplatelet drugs (*n* = 192; 27.4%), anticoagulants (*n* = 49; 7%), hypoglycemic drugs (*n* = 101; 14.4%), antihypertensive (*n* = 238; 34%), lipid-lowering (*n* = 101; 14.4%), sedatives (*n* = 81; 11.6%), and others (*n* = 255; 36.4%). Pharmacotherapy was significantly more common in the older age group in comparison to the younger (*n* = 189; 91.3% vs. *n* = 275; 56.1%; *p* < 0.0001) and mainly applied to people with chronic diseases (*n* = 459; 98.9%).

Two hundred and thirty (32.8%) of the respondents used the advice of a primary care physician, while 360 (51.4%) consulted specialist doctors (Table 1). It is worth adding that senior doctors underwent health condition monitoring significantly more frequently, with the exception of dental visits. In this age group (50–80), preventive actions were significantly more frequent, but those people also required sick leave at a significantly higher rate.

In the group of doctors from the Silesian Voivodeship (*N* = 701), 17.8% (*n* = 125) reported smoking, and 54.4% (*n* = 381) reported alcohol consumption. Almost 3/4 of the respondents (*n* = 497; 70.9%) declared the practice of active sports, while only 11.6% (*n* = 81) practiced sports regularly. Almost one-third of the respondents (*n* = 200; 28.5%) used a diet at least once in their life.

According to the HADS, the occurrence of anxiety symptoms was reported by 181 (25.8%) doctors, while depressive symptoms were reported by 135 (19.3%) respondents, and 115 doctors (16.4%) had both symptoms simultaneously. The detailed results of the HADS questionnaire are presented in Figure 2.

The severe intensification of anxiety symptoms affected the elderly (*n* = 79; 16.1% vs. *n* = 102; 48.8%; *p* < 0.0001) significantly more frequently, and this was similar for severe depressive symptoms (*n* = 49; 10.0% vs. *n* = 86; 41.1%; *p* < 0.0001). Therefore, the presented results of the analysis were considered for separate age groups. A significantly higher risk of anxiety symptoms was observed among overweight, obese, married, divorced, and widowed people living in houses. On the other hand, a good economic status was associated with a reduced risk of developing anxiety symptoms. Similar observations were observed for the risk of depressive symptoms (Table 2). Then, the following factors related to the working environment appeared to be significant for the increased risk of anxiety/depressive disorders: having a basic specialization (mainly in the younger age group), having an additional specialization, having a specialized medical practice, and a longer period of professional work (Table 3). A lower risk was related to working in a hospital, a smaller weekly workload, the type of rest, and a higher sense of job satisfaction. The severity of anxiety/depressive symptoms was also associated with the occurrence of a chronic disease, pharmacotherapy, diet, and chronic fatigue. On the contrary, frequent physical activity, a longer sleep duration, and a higher level of life satisfaction reduced the risk of developing disorders (Table 4).

## 4. Discussion

Work in health care is associated with many threats of a physical, chemical, and biological nature, and they may affect the physical or mental health of employees [16]. The physical risk factors include activities leading to diseases of the musculoskeletal system related to patient service or the positions of employees due to their professional activities. Contact with chemicals, including medicines and cleaning agents, may cause irritation of skin, eyes, or respiratory system. Medical personnel are also particularly vulnerable to infectious diseases transmitted by air or through contact with contaminated blood. The aforementioned risk factors, as well as exposure to ionizing radiation, related to the operation of X-ray machines, fluoroscopes, or computed tomographs (CT), may also lead to disorders related to the reproductive system, fertility, and fetal development [16]. Stress leading to overweight or obesity and other metabolic, heart, and circulatory system diseases, as well as depression, is also frequently mentioned [4,5,6,16]. Therefore, it is important to monitor the physical and mental health of medical doctors and their pro-health activities in relation to the selected socio-economic and professional aspects of life, as well as life and job satisfaction.

The percentages of the most frequent diseases and chronic symptoms among doctors, on the basis of the authors’ study, reached higher levels than those presented for adults in a report on the health of the Polish population in 2019 [17]. The order of prevalence was as follows: arterial hypertension (33.8% vs. 26.5%), coronary artery disease (24.4% vs. 7.5%), dyslipidemia (27.1% vs. 9.4%), allergies (26.7% vs. 8.4%), prior myocardial infarction (14.1% vs. 6.6%), and diabetes (14.1% vs. 8.1%). The percentages of doctors with arterial hypertension (33.8%) and dyslipidemia (27.1%) were also higher than the results presented by Sobrino et al., being 23.9% and 16.4%, respectively, for a group of 485 Spanish health care workers [18]. It is disturbing that almost 3/4 of the surveyed doctors from the Silesian Voivodeship had a body mass index indicating overweight or obesity, with 7% indicating obesity. Meanwhile, in the Polish population, the problem of overweight or obesity affects 57% of individuals [17]. This observation is confirmed by the results of studies conducted in the USA, Mexico, and Nigeria, according to which the rates of overweight and obesity among health care workers are significantly higher in relation to the general population [16]. It is worth adding that the incidence of chronic diseases and symptoms among the Silesian doctors was almost two times higher in the older age group. A similar prevalence profile was recorded for the Polish population in general in 2019, with arterial hypertension predominating among the group of people aged 60+ [17].

An important factor that may indirectly negatively affect doctors’ health is long working hours, which may increase the risk of bad habits such as smoking, drinking alcohol, drug abuse (especially sleeping pills), improper diet, or lack of physical activity [19,20]. The own study presented an unfavorable image of the doctors’ lifestyle, as more than half of the respondents declared alcohol consumption, while 1/5 of the respondents reported nicotinism and almost 90% of the respondents reported low physical activity. In addition, pharmacotherapy was used by over 60% of the doctors and over 10% of those who took sedatives. Unfortunately, the research conducted on groups of students of medicine has revealed numerous shortcomings in this area, including risky alcohol consumption [6] and reduced physical activity [21].

The literature focuses mainly on aspects of doctors’ mental health. The stress resulting from the peculiarity of this profession is the most important factor [16]. A study conducted on employees of the anesthesiology and intensive care team in 2012 in Łódź revealed the presence of severe stress in the emergency situations, experienced by most of the respondents (69%) [5]. In total, 24% of the respondents felt stress constantly, and only 7% did not experience stress at work. It was also observed that the number of years in work was related to the acquisition of experience and the ability to cope with stress (52% of respondents). Moreover, half of the surveyed medical staff declared that they did not transfer the stress they felt at work to their personal life (47%) or tried to minimize it (42%). The most common causes of stress were difficult intubation (59%), bad work organization (40%), resuscitation (38%), awareness of responsibility and risk (37%), and a limited number of staff (31%). Additionally, the medical stressors included non-specific patient symptoms, dyspnea, vomiting, pain, and a lack of response to the treatment [22]. Moreover, additional stress appeared when facing existential questions asked by patients and their families and the necessity to present tragic information [23], which caused feelings of helplessness, guilt, and a lack of faith. The study conducted on 411 doctors, including specialists (cardiology, hemodynamics, gynecology and obstetrics, dentistry, surgery, oncology, pediatrics, neurosurgery, psychiatry, family medicine, emergency medical services), revealed a correlation between working more than one job and experiencing occupational stress. Significant influences of the lack of reward at work (*R* = 0.14, *p* < 0.001), the lack of social contacts (*R* = 0.12, *p* < 0.01), unpleasant working conditions (*R* = 0.10, *p* < 0.05), and the lack of social support (*R* = 0.10, *p* < 0.05) in increasing occupational stress were confirmed. Additionally, the importance of having a family, as a factor limiting the effects of occupational stress, was pointed out [24].

In the authors’ study, the occurrence of anxiety symptoms affected 1/4 of the examined doctors from the Silesian Voivodeship, while the occurrence of depressive symptoms was observed in 1/5 of the respondents. Data on the health condition of the Polish population showed that 16.1% have symptoms which may indicate depression [17]. According to the review of the 2020 literature on the mental health of doctors and medical students, the symptoms of depression affected 28.8% of residents and 27.2% of medical students [6], which was consistent with our results. In our study, the risk of developing anxiety symptoms was nearly 5 times higher and the risk of depression symptoms was 6 times higher in the older age group. The higher risks of anxiety and depression were related to social and occupational roles, lower economic status, overweight or obesity, chronic diseases, pharmacotherapy, diet, and chronic fatigue. On the other hand, frequent physical activity, a longer sleep duration, lower weekly workload, type of rest, a higher sense of job satisfaction, and a higher satisfaction with life reduced the risk of disorders. The present analyses revealed two groups of factors influencing the occurrence of anxiety and depressive symptoms. The so-called soft indicators, including job satisfaction and life satisfaction, determined subjectively, were presented among the protective factors. There was a surprising observation that having specialization and/or a private medical practice increased the risk of developing anxiety symptoms. On the one hand, these elements increase the socio-economic status of an individual and, therefore, should engender not only prestige but also a sense of economic safety. On the other hand, a higher economic status may affect the psychological burden associated with a greater level of care for the possessed goods and maintaining the achieved social position. A cross-sectional study of 150,000 people from 26 countries showed that anxiety disorder was more prevalent and disabling in high-income countries than in low- and middle-income countries [25]. One of the theories explaining these dependencies indicated that individual worrying tendencies may be more prevalent under the conditions of relative wealth and stability [25]. In low- and middle-income countries, economic disadvantage appears to be permanent, and individuals and groups have adapted to it; thus, it does not adversely affect mental health. However, in the examined group, this explanation does not seem to apply, as the doctors can be considered as belonging to one economic group. Many studies also indicated the relationship between personality and health, and, perhaps, in this area, an attempt should be made to explain the observed differences [26,27]. The Five Factor Model (FFM) measuring personality in relation to the main characteristics of openness, conscientiousness, extraversion, agreeableness, and neuroticism is particularly applicable [28]. The research indicated that the features of conscientiousness had a positive impact on the health of adults and were associated with a lower risk of mortality [26]. The conducted study did not analyze the personality traits of the surveyed doctors, but it is possible that, in addition to the socio-economic factors, these traits were also important determinants of the health condition (this may encourage further studies on this topic). The studies of other authors highlighted the relationship between the occurrence of depressive disorders and a high level of neuroticism, as well as the use of hypnotics and sedatives [6].

While individual physical and mental well-being and chronic stress were associated with job satisfaction and even general life satisfaction, the present overview of the doctors’ health condition implies that the occurrence of burnout in this professional group should also be mentioned [27]. This work-related erosion syndrome appears in the medical profession regardless of specialization, but the palliative care units, the oncology units, and other units treating diseases with a high mortality rate are mentioned particularly frequently. The psychological analysis of this condition leads to a strong awareness of finiteness and mortality among doctors, and the inability to maintain an appropriate emotional distance from pain and suffering. It is common for workplace situations to prompt health professionals to rethink their lives, relationships, and activities, which, in some cases, can be problematic and result in depression, rebellion, or retreat from work. That is why the ability to accept physical and mental suffering plays such an important role in this context. A survey conducted in 2014 on the medical staff of the hospice in Poznan revealed a high level of professional commitment (54% of respondents), despite the difficult conditions in the workplace [22]. As many as 94% of respondents did not feel discouraged from going to work. Despite the high percentage of respondents expressing satisfaction with their profession and workplace, as many as 84% of the respondents confirmed the occurrence of depression and felt anxiety caused by work. Most of the respondents (84%) expressed the opinion that neglecting rest, excessive dedication to work, and the lack of a personal life were the main factors influencing the occurrence of the burnout syndrome. Despite frequent anxiety and depression resulting from their professional duties, none of the respondents expressed the need to change their place of work. This is possibly due to the fact that the majority of the examined people were able to separate their private life from their professional life (62%) but, unfortunately, 42% did not know how to proceed in the case of the burnout syndrome.

Maintaining physical and mental balance in the medical profession is a difficult and complex challenge. It is important in this process to set priorities and to balance activities and emotional reactions between work and non-working life. The importance of planning days devoted to rest and contact with other people, including those outside one’s professional circles, has been emphasized [29]. In addition, the results of research conducted in this area allow us to provide the following suggestion: information about the causes of burnout and methods of dealing with it should be introduced at the stage of preparation for the medical profession [24,30,31].

The authors of the study are aware that the results do not fully reflect the current health situation, as the data originate from the penultimate year before the COVID-19 pandemic. However, the results can provide a basis for further comparisons and analysis and can offer a valuable reference point for research on the impact of a pandemic on the health of doctors.

It is also worth discussing the main limitation of the study, which is the research model adopted (cross-sectional study). It does not allow us to draw cause-and-effect conclusions and presents only the frequency rates of health problems in the examined group of medical doctors and the potential risk factors. However, it is an important source of information that can offer a basis for the construction of preventive measures. Moreover, the structural consistency of the study group, with respect to the population in the present study, was ensured, taking into account the demographic (age groups, gender) and the professional (specialization) representation.

Another limitation is the assessment of the frequency of emotional disorders and the level of satisfaction with life or work based on self-assessment questionnaires. Hence, it is important to use standardized and validated questionnaires, such as the HADS [13], the SWLS [14], and job satisfaction questionnaires [15]. Despite the use of common research tools, it is difficult to discuss the results in relation to the available studies from other countries. This is due to significant differences in health systems and economic and social conditions.

Taking into account the picture of the health condition of Polish medical doctors presented here, certain preventive measures should be taken. The WHO study addressing employers and employees of developing countries seems to be valid for the health care sector in Poland with respect to efforts to reduce work-related stress [32]. The main stress factors related to the work environment (fast work pace, time pressure, long working hours, low income), home environment (conflict of responsibilities and roles, particularly for women, difficulties in daily life logistics), and self-confidence are common to every work environment. Therefore, detecting the signs of work-related stress, analyzing the risk factors and the risk groups, and undertaking preparatory actions, e.g., raising awareness about work-related stress and the importance of balancing work and personal life, are the starting points for improving the situation. It seems reasonable to supplement medical students’ lectures with psychological workshops in order to teach them how to cope with stress, among other challenges. With regard to working medical doctors, conducting psychological workshops may be an ineffective method. It may have the opposite effect, increasing frustration and increasing the stress associated with the need to undertake this type of activity. Mental health prophylaxis and the practice of work–life balance among medical doctors should be considered holistically and in a broad context. It is not enough to indicate only the possible causes of mental health problems or worse quality of health in the analyzed group. It is also necessary to consider the causes of the causes of this condition. In Poland, this problem may have a systemic dimension. The insufficient number of specialists, the outflow of young doctors from Poland, and their practice of taking up work in Western Europe may increase staffing problems and the need to work above the standard hours and in several places. Thus, this situation is not conducive to work–life balance and only increases stress and possible non-work problems. Providing fast and appropriate mental health services for the population of medical doctors is also important. While, at the level of the medical unit, measures based on evaluations and support systems may prove—as already mentioned—that improvements in the Polish health care system are absolutely necessary. The lack of specialists in the field of mental health is a serious national problem, and this makes it impossible to act quickly, which also applies to this particularly vulnerable occupational group. The demographic condition of working doctors in Poland is also worrisome, as 1/3 people are at the pre-retirement or retirement age. This group is naturally prone to chronic diseases. A limited number of specialists and a relatively low salary results in additional working time, which is linked to general fatigue, lowered mental and physical conditions, interpersonal conflicts, and neglect of a healthy lifestyle. Therefore, it is necessary to strengthen and improve the public health system in Poland, starting with the stage of young doctors’ education and extending to changes in the financing policy and the integration of health care subsystems. This study additionally points to the broad dimensions of health and its determinants. With regard to the mental health of medical doctors working in Poland, the important elements are macro-social factors, the health care system, staff limitations, size of the salary, the working conditions, and the migration of young doctors. All these elements create an interconnected whole, which probably reflects cause-and-effect relationships and affects the mental health of this occupational group. Thus, mental health prophylaxis is a significant challenge not only for people dealing with prevention but also for health care decision makers.

## 5. Conclusions

Doctors from the Silesian Voivodeship most often reported cardiovascular diseases (hypertension, coronary artery disease, myocardial infarction, atrial fibrillation) and endocrine diseases, nutritional status and metabolic changes (overweight or obesity, diabetes, dyslipidemia), allergies, and degenerative changes in the musculoskeletal system and connective tissue. The chronic diseases, as well as anxiety and depressive disorders, more often affected doctors aged 50–80, who were also more likely to undertake pro-health activities. The higher risks of anxiety and depression were related to the individual’s social and professional role, lower economic status, overweight or obesity, chronic disease, pharmacotherapy, diet, and chronic fatigue. On the other hand, frequent physical activity, a longer sleep duration, lower weekly workload, type of rest, a higher sense of job satisfaction, and a higher level of satisfaction with life reduced the risk of disorders. The health condition of doctors is unfavorably different from the psychophysical condition of the general Polish population and requires decisive remedial actions in the professional and non-professional spheres. At the health care system level, changes are needed in the following areas: the education of young doctors, changes in the financial policy, and the integration of health care subsystems. At the occupational level, actions related to evaluation and support systems are also essential. There is also a necessity for personal preventive measures aimed at maintaining a healthy lifestyle, controlling one’s health, and reducing stress.

## Figures and Tables

**Figure 1 ijerph-19-13410-f001:**
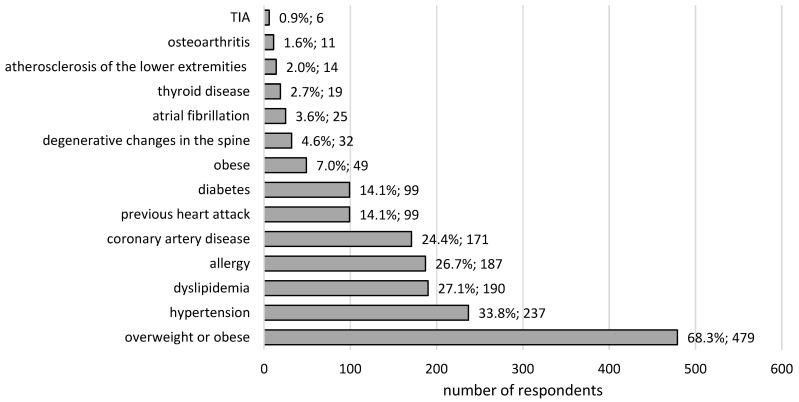
The most common diseases in the group of surveyed doctors (N = 701; 100%).

**Figure 2 ijerph-19-13410-f002:**
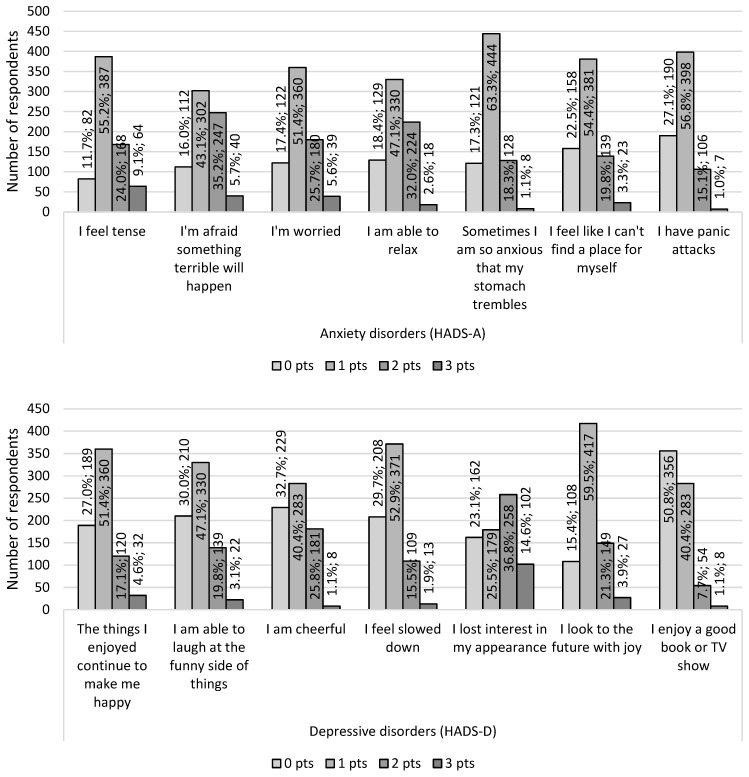
HADS scale results regarding the occurrence of anxiety symptoms (HADS-A) and depressive symptoms (HADS-D) in the group of examined doctors (N = 701; 100%).

**Table 1 ijerph-19-13410-t001:** Data on health control in the group of examined doctors.

Health Care		Total N = 701 (100%)	25–50 N = 492 (100%)	50–80 N = 209 (100%)	*p*-Value
GP control visits n (%)	no	52 (7.4)	35 (7.1)	17 (8.1)	<0.001
	I do not remember	419 (59.8)	316 (64.2)	103 (49.3)	
	yes	230 (32.8)	141 (28.7)	89 (42.6)	
GP doctor—last visit (years)	X ± S M (IQR)	2.3 ± 1.8 2.0 (2.0)	2.6 ± 2.0 2.0 (2.0)	1.9 ± 1.5 1.0 (1.0)	<0.01
Specialist control visits n (%)	never	315 (44.9)	285 (58.0)	30 (14.3)	<0.0001
	I do not remember	26 (3.7)	19 (3.9)	7 (3.4)	
	yes	360 (51.4)	188 (38.3)	172 (82.3)	
Specialist—last visit (years)	X ± S M (IQR)	2.1 ± 1.8 2.0 (1.0)	2.4 ± 1.9 2.0 (2.0)	1.8 ± 1.6 1.0 (1.0)	<0.0001
Dentist n (%)		697 (99.4)	490 (99.6)	207 (99.0)	0.74
Dentist—last visit (years)	X ± S M (IQR)	1.8 ± 1.0 2.0 (1.0)	1.7 ± 0.8 2.0 (1.0)	2.0 ± 1.3 2.0 (1.0)	0.04
Gynecologist—N_w_ = 336/243/93 (100%) n (%)		323 (96.1)	231 (95.1)	92 (98.9)	0.18
Gynecologist—last visit (years)	X ± S M (IQR)	2.1 ± 1.5 2.0 (2.0)	2.1 ± 1.3 2.0 (2.0)	2.3 ± 1.8 2.0 (2.0)	0.68
Urologist—N_M_ = 365/249/115 (100%) n (%)		63 (17.3)	22 (8.8)	41 (35.7)	<0.0001
Urologist—last visit (years)	X ± S M (IQR)	2.6 ± 1.7 2.0 (1.0)	2.4 ± 1.3 2.0 (1.0)	2.7 ± 1.9 2.0 (1.0)	0.59
Glasses/lenses n (%)		498 (71)	320 (65)	178 (85.2)	<0.0001
Blood pressure control n (%)	I do not remember	16 (2.3)	14 (2.6)	2 (1.0)	<0.0001
	a few years ago	8 (1.1)	7 (1.4)	1 (0.5)	
	1–2 years ago	49 (7.0)	48 (9.8)	1 (0.5)	
	several times a year	290 (41.4)	267 (54.3)	23 (11.0)	
	several times a month	253 (36.1)	133 (27.2)	120 (57.4)	
	a few times a week	71 (10.1)	21 (4.3)	50 (23.9)	
	every day	14 (2.0)	2 (0.4)	12 (5.7)	
Glucose control n (%)	never	5 (0.7)	5 (1.0)	0 (0.0)	<0.0001
	I do not remember	18 (2.6)	15 (3.0)	3 (1.4)	
	a few years ago	113 (16.1)	106 (21.5)	7 (3.3)	
	1–2 years ago	302 (43.1)	240 (48.8)	62 (29.7)	
	in the last year	263 (37.5)	126 (25.6)	137 (65.6)	
Lipid control n (%)	never	16 (2.3)	15 (3.0)	1 (0.6)	<0.0001
	I do not remember	26 (3.7)	23 (4.7)	3 (1.4)	
	a few years ago	104 (14.8)	101 (20.5)	3 (1.4)	
	1–2 years ago	292 (41.7)	222 (45.1)	70 (33.4)	
	in the last year	263 (37.5)	131 (26.6)	132 (63.2)	
Densitometry n (%)		104 (14.8)	18 (3.7)	49 (23.4)	<0.0001
Sick leave n (%)		596 (85.0)	394 (80.1)	202 (96.7)	<0.0001
Sick leave (number of days in the year)	X ± S M (IQR)	16.7 ± 24.1 10.0 (10.0)	13.0 ± 20.4 7.0 (9.0)	23.9 ± 28.7 14.0 (20.0)	<0.0001
Sick leave—term n (%)	never	104 (14.8)	97 (19.8)	7 (3.3)	<0.0001
	over 5 years ago	98 (14.0)	69 (14.1)	29 (13.9)	
	3–5 years ago	152 (21.7)	121 (24.6)	31 (14.8)	
	1–2 years ago	195 (27.8)	135 (27.5)	60 (28.7)	
	in the last year	151 (21.5)	69 (14.1)	82 (39.2)	

GP—general practitioner; N_W_—number of women; N_M_—number of men; 25–50—age group 25 years old or more and less than 50 years old; 50–80—age group 50 years old or more and less than 80 years old. Measurable data presented as mean X and standard deviation S and median M and quartile range IQR; *p*-value according to Student’s *t*-test or the Mann–Whitney U test. Nominal data presented by the quantitative percentage notation n (%); *p*-value according to the X2 test or the Fisher test.

**Table 2 ijerph-19-13410-t002:** Crude odds ratios (OR) with 95% confidence interval (CI) for the relationship between the occurrence of anxiety disorders (12 points or more on the HADS-A scale) and depression (12 points or more on the HADS-D scale) and anthropometric and socio-economic factors.

Anthropometric Data/ Social/ Economic: Reference Group		HADS-A						HADS-D					
		Total		25–50		50–80		Total		25–50		50–80	
		OR [95% CI]	*p*-Value	OR [95% CI]	*p*-Value	OR [95% CI]	*p*-Value	OR [95% CI]	*p*-Value	OR [95% CI]	*p*-Value	OR [95% CI]	*p*-Value
Age: 25–50 years old	50–80 age group	4.98 [3.47–7.16]	<0.0001	-	-	-	-	6.32 [4.22–9.47]	<0.0001	-	-	-	-
Sex: women	men	0.76 [0.54–1.07]	0.11	0.54 [0.33–0.89]	0.02	0.88 [0.51–1.52]	0.65	1.38 [0.94–2.02]	0.10	1.02 [0.56–1.84]	0.95	1.66 [0.95–2.91]	0.08
BMI: norm	underweight	1.10 [0.12–10.11]	0.93	1.90 [0.20–17.86]	0.57	-	-	-	-	-	-	-	-
	overweight	1.68 [1.13–2.52]	0.01	1.63 [0.95–2.83]	0.08	0.74 [0.36–1.56]	0.44	2.63 [1.59–4.36]	<0.001	3.74 [1.63–8.58]	<0.01	0.98 [0.46–2.08]	0.95
	obesity	3.30 [1.70–6.40]	<0.001	3.56 [1.30–9.72]	0.01	0.91 [0.33–2.48]	0.85	5.88 [2.83–12.20]	<0.0001	7.31 [2.09–25.54]	<0.01	1.62 [0.59–4.45]	0.35
Family status: free status	in relation with partner	0.69 [0.29–1.66]	0.41	0.72 [0.28–1.85]	0.08	0.22 [0.02–2.97]	0.26	1.97 [0.77–5.04]	0.16	2.49 [0.89–7.01]	0.08	0.22 [0.02–2.97]	0.26
	married	2.64 [1.66–4.20]	<0.0001	1.48 [0.86–2.56]	0.59	1.25 [0.27–5.77]	0.78	4.57 [2.43–8.60]	<0.0001	2.76 [1.27–6.01]	0.01	0.88 [0.19–4.09]	0.87
	divorced	4.34 [2.17–8.70]	<0.0001	2.93 [1.15–7.47]	0.02	1.58 [0.29–8.61]	0.60	6.32 [2.73–14.66]	<0.0001	4.58 [1.40–14.94]	0.01	0.95 [0.17–5.23]	0.96
	widowed	9.43 [3.36–26.45]	<0.0001	-	-	2.10 [0.36–12.32]	0.41	23.18 [7.61–70.58]	<0.0001	-	-	2.10 [0.36–12.32]	0.41
Children: no	yes	1.83 [0.82–4.09]	0.14	1.88 [1.14–3.11]	0.01	0.82 [0.29–2.36]	0.72	4.02 [2.43–6.66]	<0.0001	2.86 [1.48–5.54]	<0.01	0.79 [0.27–2.25]	0.65
Partner working in the medical profession: no	yes	1.22 [0.82–1.81]	0.33	1.15 [0.63–2.10]	0.65	1.20 [0.67–2.16]	0.54	1.19 [0.78–1.82]	0.43	1.31 [0.66–2.62]	0.44	1.03 [0.56–1.86]	0.93
Subjective assessment of economic status: bad	medium	0.58 [0.29–1.16]	0.12	0.45 [0.19–1.04]	0.06	0.20 [0.02–1.69]	0.14	0.69 [0.33–1.44]	0.32	0.50 [0.19–1.3]	0.16	0.35 [0.07–1.88]	0.22
	good	0.28 [0.12–0.64]	<0.01	0.29 [0.10–0.84]	0.02	0.05 [0.01–0.50]	0.01	0.24 [0.09–0.62]	<0.01	0.11 [0.02–0.60]	0.01	0.10 [0.02–0.60]	0.01
	no opinion	0.34 [0.14–0.80]	0.01	0.41 [0.15–1.14]	0.09	0.22 [0.02–2.97]	0.26	0.26 [0.10–0.72]	0.01	0.36 [0.11–1.22]	0.10	0.16 [0.02–1.63]	0.12
Place of residence: flat	house	2.64 [1.86–3.74]	<0.0001	1.20 [0.66–2.16]	0.55	1.37 [0.75–2.53]	0.31	3.53 [2.40–5.21]	<0.0001	1.81 [0.93–3.53]	0.08	1.48 [0.79–2.77]	0.23
Credit obligations: no	yes	0.91 [0.65–1.27]	0.58	1.10 [0.68–1.79]	0.69	0.77 [0.45–1.33]	0.35	0.73 [0.50–1.07]	0.11	0.93 [0.51–1.68]	0.81	0.61 [0.35–1.07]	0.09

OR (95% CI)—crude odds ratio with a 95% confidence interval; *p*-value according to the test of statistical significance; 25–50—age group 25 years old or more and less than 50 years old; 50–80—age group 50 years old or more and less than 80 years old.

**Table 3 ijerph-19-13410-t003:** Crude odds ratios (OR) with 95% confidence interval (CI) for the relationship between the occurrence of anxiety disorders (12 points or more on the HADS-A scale) and depression (12 points or more on the HADS-D scale) and occupational factors.

Professional Environment: Reference Group		HADS-A						HADS-D					
		Total		25–50		50–80		Total		25–50		50–80	
		OR [95% CI]	*p*-Value	OR [95% CI]	*p*-Value	OR [95% CI]	*p*-Value	OR [95% CI]	*p*-Value	OR [95% CI]	*p*-Value	OR [95% CI]	*p*-Value
Primary specialization: no	yes	3.82 [1.95–7.51]	<0.0001	2.06 [1.02–4.16]	0.04	-	-	9.67 [3.02–30.99]	<0.0001	4.59 [1.40–15.06]	0.01	-	-
Additional specialization: no	yes	1.65 [1.11–2.47]	0.01	1.36 [0.69–2.70]	0.38	0.78 [0.44–1.36]	0.38	1.65 [1.07–2.56]	0.02	2.02 [0.95–4.29]	0.07	0.59 [0.33–1.06]	0.08
Academic degree: no	yes	1.20 [0.83–1.75]	0.33	1.98 [1.21–3.25]	0.01	1.06 [0.52–2.14]	0.87	1.06 [0.69–1.61]	0.80	2.05 [1.13–3.73]	0.02	0.92 [0.45–1.88]	0.82
Hospital: no	yes	0.28 [0.19–0.40]	<0.0001	0.46 [0.24–0.86]	0.02	0.52 [0.30–0.91]	0.02	0.23 [0.15–0.34]	<0.0001	0.32 [0.16–0.65]	<0.01	0.53 [0.30–0.93]	0.03
GP clinic: no	yes	1.07 [0.71–1.61]	0.75	0.86 [0.50–1.48]	0.59	0.82 [0.38–1.73]	0.60	1.26 [0.79–2.02]	0.33	0.94 [0.48–1.83]	0.85	1.03 [0.48–2.21]	0.95
Having a specialized clinic: no	yes	2.48 [1.75–3.51]	<0.0001	2.18 [1.34–3.54]	0.002	1.08 [0.60–1.96]	0.79	2.85 [1.92–4.23]	<0.0001	2.76 [1.51–5.03]	<0.001	1.10 [0.60–2.02]	0.75
Work period (years)		1.08 [1.07–1.10]	<0.0001	1.10 [1.06–1.14]	<0.0001	1.06 [1.01–1.11]	0.01	1.09 [1.07–1.11]	<0.0001	1.14 [1.09–1.20]	<0.0001	1.04 [1.00–1.09]	0.06
Number of working hours (hour/week)		0.98 [0.96–0.99]	<0.01	0.99 [0.97–1.02]	0.54	0.99 [0.97–1.00]	0.14	0.98 [0.96–0.99]	<0.01	1.00 [0.96–1.03]	0.78	0.99 [0.97–1.00]	0.12
Post-shift time at the primary workplace: resting	working	0.99 [0.66–1.47]	0.95	1.28 [0.77–2.13]	0.34	0.44 [0.21–0.91]	0.03	0.82 [0.52–1.30]	0.41	0.99 [0.53–1.86]	0.98	0.45 [0.22–0.96]	0.04
Place of rest: in the country	abroad	0.35 [0.17–0.75]	0.01	0.55 [0.20–1.52]	0.25	0.24 [0.07–0.77]	0.02	0.35 [0.15–0.81]	0.01	0.49 [0.14–1.70]	0.26	2.56 [1.34–4.87]	<0.01
	in the country and abroad	0.35 [0.24–0.50]	<0.0001	0.44 [0.26–0.73]	<0.01	0.53 [0.29–0.98]	0.04	0.25 [0.17–0.39]	<0.0001	0.33 [0.18–0.62]	<0.001	0.79 [0.23–2.73]	0.71
Annual sick leave (days)		0.95 [0.91–1.00]	0.03	0.93 [0.88–1.00]	0.04	1.00 [0.94–1.06]	0.99	0.96 [0.92–1.01]	0.10	0.94 [0.87–1.02]	0.14	1.00 [0.94–1.07]	0.92
Leisure trips (N/year)		0.99 [0.91–1.08]	0.85	1.08 [0.96–1.21]	0.19	0.64 [0.48–0.85]	<0.01	0.56 [0.45–0.71]	<0.0001	0.70 [0.5–0.99]	0.04	0.61 [0.45–0.83]	<0.01
Professional satisfaction (points)		0.94 [0.92–0.96]	<0.0001	0.95 [0.92–0.98]	<0.001	0.94 [0.91–0.97]	<0.001	0.93 [0.91–0.95]	<0.0001	0.94 [0.91–0.97]	<0.001	0.93 [0.89–0.96]	<0.0001

OR (95% CI)—crude odds ratio with a 95% confidence interval; *p*-value by the test of statistical significance; 25–50—age group 25 years old or more and less than 50 years old; 50–80—age group 50 years old or more and less than 80 years old.

**Table 4 ijerph-19-13410-t004:** Crude odds ratios (OR) with 95% confidence interval (CI) for the relationship between the occurrence of anxiety disorders (12 points or more on the HADS-A scale) and depression (12 points or more on the HADS-D scale) and health status or lifestyle.

Health/ Lifestyle: Reference Group		HADS-A						HADS-D					
		Total		25–50		50–80		Total		25–50		50–80	
		OR [95% CI]	*p*-Value	OR [95% CI]	*p*-Value	OR [95% CI]	*p*-Value	OR [95% CI]	*p*-Value	OR [95% CI]	*p*-Value	OR [95% CI]	*p*-Value
Chronic disease: no	yes	5.80 [3.46–9.73]	<0.0001	3.14 [1.77–5.55]	<0.0001	12.76 [1.63–100.01]	0.02	7.43 [3.82–14.46]	<0.0001	3.62 [1.72–7.65]	<0.001	9.19 [1.17–72.06]	0.03
Pharmacotherapy: no	yes	5.26 [3.23–8.57]	<0.0001	3.05 [1.74–5.35]	<0.0001	5.5 [1.54–19.62]	<0.01	7.28 [3.84–13.80]	<0.0001	4.4 [2.02–9.62]	<0.001	3.92 [1.10–13.98]	0.04
Sick leave (days/year)		1.01 [1.00–1.01]	0.09	1.00 [0.99–1.01]	0.84	1.00 [0.99–1.01]	0.84	1.01 [1.00–1.01]	0.06	1.00 [0.99–1.02]	0.98	1.00 [0.99–1.01]	0.84
Physical activity: irregular	frequent	0.19 [0.07–0.53]	<0.01	0.32 [0.10–1.06]	0.06	0.07 [0.01–0.53]	0.01	0.14 [0.03–0.58]	0.01	0.18 [0.02–1.32]	0.09	0.10 [0.01–0.82]	0.03
Diet: never	yes whenever	3.33 [2.33–4.77]	<0.0001	1.95 [1.12–3.36]	0.02	2.71 [1.55–4.74]	<0.001	4.41 [2.98–6.54]	<0.0001	3.06 [1.64–5.72]	<0.001	2.82 [1.59–4.99]	<0.001
Alcohol: no	yes	0.78 [0.55–1.09]	0.15	0.58 [0.36–0.94]	0.03	1.07 [0.62–1.84]	0.81	0.99 [0.68–1.44]	0.94	0.86 [0.47–1.55]	0.61	1.13 [0.65–1.96]	0.67
Chronic fatigue: no	yes	7.98 [5.47–11.64]	<0.0001	6.27 [3.75–10.48]	<0.0001	5.23 [2.81–9.73]	<0.0001	5.93 [3.95–8.92]	<0.0001	2.86 [1.55–5.27]	<0.001	5.30 [2.74–10.26]	<0.0001
Sleep (hours)		0.71 [0.56–0.90]	<0.01	0.68 [0.48–0.97]	0.03	1.16 [0.81–1.68]	0.42	0.74 [0.57–0.96]	0.02	0.74 [0.48–1.14]	0.18	1.16 [0.80–1.68]	0.44
Smoking: no	yes	1.26 [0.82–1.93]	0.29	1.30 [0.72–2.35]	0.39	1.39 [0.67–2.86]	0.37	1.50 [0.95–2.37]	0.08	1.96 [1.01–3.83]	0.05	1.35 [0.66–2.78]	0.42
Packet Years of Smoking		1.04 [1.01–1.08]	0.01	1.03 [0.97–1.09]	0.30	1.01 [0.96–1.06]	0.82	1.03 [0.99–1.06]	0.12	1.04 [0.98–1.10]	0.19	0.96 [0.91–1.02]	0.17
Life satisfaction (points)		0.77 [0.73–0.82]	<0.0001	0.76 [0.70–0.83]	<0.0001	0.80 [0.74–0.87]	<0.0001	0.81 [0.77–0.86]	<0.0001	0.84 [0.77–0.91]	<0.0001	0.83 [0.76–0.89]	<0.0001

OR (95% CI)—crude odds ratio with a 95% confidence interval; *p*-value by the test of statistical significance; 25–50—age group 25 years old or more and less than 50 years old; 50–80—age group 50 years old or more and less than 80 years old.

## Data Availability

The data presented in this study are available on request from the corresponding author. The data are not publicly available due to privacy reasons.

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
