# Peer review of "The Physical and Mental Well-Being of Medical Doctors in the Silesian Voivodeship"

_ijerph, 2022, doi:10.3390/ijerph192013410_

Round 1

Reviewer 1 Report

The paper focuses on a very relevant topic, with important social and political namely in what concerns health policies, as the heavy burden carried by the doctors is a worldwide problem. This study also allows for understanding the characteristics of doctors’ mental and physical health problems in Poland, which emphasizes its cultural relevance.

However, it requires a revision concerning both its structure and content. Otherwise, its publication will not be feasible. Some grammatical revision is also needed.

In the abstract, the sentence regarding the dimensions assessed and the instruments used is not completely clear.

In the introduction, in the second paragraph of page 2 (lines 52-61), adding some evidence on the prevalence of psychopathology in doctors will possibly make this section clearer and robust.

The materials and methods’ section must be revised regarding its structure and content, as it is confusing and incomplete. The inclusion of subsections regarding participants, measures, procedures and analytic plan will be helpful to improve this section organization and allow for completing some missing information. The participants’ description should be more detailed and include the mean and standard deviation for the participants’ age. To add information on sociodemographic and professional issues to characterize the sample will be helpful. The target population, the dimensions assessed, the structure, the scores and the psychometric properties of each instruments used should be described in separate subsections of the measures’ section. Data collection procedures require a more extensive description.

In what concerns the results section, in line 117, please note that the ‘230’ should be spelled out, and not in numerals. The p value should be always italicized. In table 1, the NS acronym is not necessary, as the authors present the significance level values both in the text and tables. In table 2, the ages should be presented like this: 25.50 years (and not 25,50). Please round off  all the results to two decimal places and note that, if a value does not have the potential to exceed 1.0, the leading zero should not be used (e.g., the p value). This rule should be applied along all the description of the results. In addition, I  advise the authors to revise the guidelines to report odds ratio in the text of a manuscript (e.g., APA style).

The discussion should be restructured. In its current form, it resembles to a literature review with some notes on the study’s results, as the authors tend to describe the existing evidence and confront it with the results observed in their research, and not the opposite, as it would be expected. Also, I suggest the authors to refer the research goal in the beginning of the discussion (possibly at the end of the first paragraph), as a starting point for the reflection on your results will possibly increase the coherence of this section.

In the discussion, the authors highlight the high prevalence of anxiety and depression symptoms reported by the doctors engaged in the study, underlining the protective role of job satisfaction and life satisfaction in the development of psychopathology. They also emphasize the importance of balancing work and personal life. Nevertheless, how can this balance be promoted? What are the implications of these results for the hospitals’ policies management? What about the importance of providing fast and appropriate mental health services for this population? How do you think the promotion of pro-health activities among doctors will contribute to improve their well-being and health? What kind of promotional and preventive interventions should be useful to decrease the doctors’ physical and socioemotional problems?

Finally, practical implications are missing in the conclusion. The inclusion of the reflection on the practices which will be helpful to promote doctors’ resilience will allow the authors to highlight the clinical, social, and political significance of this work.

Author Response

First, we would like to thank for this very detailed analysis of the manuscript. We agree with the majority of the Reviewers’ comments. Our replies are written in a red. We made several changes in our manuscript believing they address all the Reviewers’ comments (also marked in red).

Reviewer 2 Report

To the authors,

The  interest in assessing the physical and mental health of Polish doctors and their pro-health activities in relation to selected socio-economic and professional aspects as well as life and job satisfaction is of interest to us. However, I have a few comments that I hope could be helpful in improving the manuscript.

 Introduction

·       The introduction focuses on the presentation of findings obtained in studies conducted with samples of Polish health professionals. It would be desirable for the authors to incorporate a few sentences giving an "overview" of previous literature on the health status (psychological and mental) of doctors in other geographical settings in order to situate the scope of the problem.

·       We consider it appropriate for the authors to comment on whether the data they provide in the introduction refers to official studies-reports conducted exclusively on the Polish population (e.g., see lines 38, 40, 48).

·       We recommend that the authors clarify the comments on lines 62-63 by indicating if, when they state "A review of the literature shows that in the analysis of the health condition of doctors, only the issues of occupational exposure, especially exposure to chronic stress, are taken into account [1,3-5]", they are referring to the review of studies conducted only with the Polish population.

·       Could the authors go beyond the statement of a general objective of their research (see lines 65-67) and detail the various sub-objectives guiding the application of the statistical tests and the results obtained?

 Materials and Methods

·       We suggest to the authors the possibility of structuring this section in different subsections. We believe that this would provide greater clarity on the research procedure.  Some of the possible subsections to include would be: Procedure and participants, measures/instruments and statistical analysis.

·       We recommend that the authors describe in greater detail the characteristics of the different self-reports applied (e.g., Hospital Anxiety and Depression Scale and Satisfaction with Life Scale). Some aspects to consider: number of items that compose them, measurement scales used, examples of items, psychometric properties.

Discussion

·       Since in this section the main risk factors and health protection factors of the sample of Polish physicians analyzed are discussed, we recommend the authors to advance in the practical implications of their research. We suggest that, on the basis of different preventive and/or interventive proposals carried out in the field, they present in greater depth measures of action that could be effective in addressing both the physical and psychological health of this professional group.

·       We recommend a detailed exposition of the main strengths and limitations of this study.

Bibliography

·       With regard to the bibliography, it is worth noting the inclusion of a large number of references to studies carried out in Poland and very few references to research carried out in other countries. From our perspective, as we have already pointed out, the inclusion of more works carried out in other latitudes would enrich the introduction and discussion of the results obtained in this research.

Author Response

(The authors gave the same response as above.)

Round 2

Reviewer 1 Report

The revision performed by the authors substantially improved the manuscript’s quality. In the introduction, the information added on the incidence of psychopathology in doctors makes this section clearer. The inclusion of a more detailed description both study’s goals, along with the changes introduced in the Materials and Methods’ section, made the methodological issues more robust. The subsections added concerning the sample, the instruments used and their composition, as well as the analytic plan, also contribute to increase the manuscript’s consistency. However, the eligibility criteria are usually part of the subsection regarding the participants/sample’s description.

In what concerns the discussion, the inclusion of the study’s limitations and suggestions for future research makes the structure of this section more coherent and consistent. However, I still think that the distinction between the study’s results and previous evidence remains unclear. To coherently present the study’s results first and confront them with the existing evidence after will possibly be a helpful strategy.

Furthermore, the authors added reflection on the practical implications of their work at the end of the conclusion, which highlights its relevance.

Therefore, this revised version of the paper meets the criteria needed for publication.